# Genome sequence and silkomics of the spindle ermine moth, *Yponomeuta cagnagella*, representing the early diverging lineage of the ditrysian Lepidoptera

Anna Volenikova [1,2,9], Petr Nguyen [1,2,9], Peter Davey[3,4], Hana Sehadova [1,2], Barbara Kludkiewicz[1], Petr Koutecky[2], James R. Walters [5], Peter Roessingh[6], Irena Provaznikova[1,2,8], Michal Sery[1], Martina Zurovcova[1], Miluse Hradilova[7], Lenka Rouhova[1,2] & Michal Zurovec [1,2✉]

Many lepidopteran species produce silk, cocoons, feeding tubes, or nests for protection from predators and parasites for caterpillars and pupae. Yet, the number of lepidopteran species whose silk composition has been studied in detail is very small, because the genes encoding the major structural silk proteins tend to be large and repetitive, making their assembly and sequence analysis difficult. Here we have analyzed the silk of *Yponomeuta cagnagella*, which represents one of the early diverging lineages of the ditrysian Lepidoptera thus improving the coverage of the order. To obtain a comprehensive list of the *Y. cagnagella silk* genes, we sequenced and assembled a draft genome using Oxford Nanopore and Illumina technologies. We used a silk-gland transcriptome and a silk proteome to identify major silk components and verified the tissue specificity of expression of individual genes. A detailed annotation of the major genes and their putative products, including their complete sequences and exon-intron structures is provided. The morphology of silk glands and fibers are also shown. This study fills an important gap in our growing understanding of the structure, evolution, and function of silk genes and provides genomic resources for future studies of the chemical ecology of *Yponomeuta* species.

[1] Biology Centre of the Czech Academy of Sciences, Institute of Entomology, Ceske Budejovice, Czech Republic. [2] Faculty of Science, University of South Bohemia, Ceske Budejovice, Czech Republic. [3] School of Natural and Environmental Sciences, Newcastle University, Newcastle upon Tyne, UK. [4] NatureMetrics Ltd, Surrey Research Park, Guildford GU2 7HJ, UK. [5] Department of Ecology and Evolutionary Biology, University of Kansas, Lawrence, USA. [6] Institute for Biodiversity and Ecosystem Dynamics, University of Amsterdam, Amsterdam, The Netherlands. [7] Institute of Molecular Genetics, Academy of Sciences of the Czech Republic, Praha, Czech Republic. [8] Present address: European Molecular Biology Laboratory, Heidelberg, Germany. [9] These authors contributed equally: Anna Volenikova, Petr Nguyen. ✉email: zurovec@entu.cas.cz

Silk is a functional term used to describe protein fibers spun by a number of arthropod lineages and encompasses a wide range of different materials[1]. The increasing application of omics technologies to characterize the nucleotide and protein sequences specific for silk components is revealing striking variability in silk properties across arthropod taxa[2–5]. It has been hypothesized that silks with different dominant protein structures have different evolutionary origins. A growing number of studies in spiders[6], moths[2,7], caddisflies[8,9] and honeybees[10,11] suggest that silk production has evolved independently in different groups. Insects use different types of glands for silk secretion and produce a range of protein secretions. Indeed, insect silk may have evolved independently in 23 lineages[1]. However, the larvae of both Lepidoptera and Trichoptera (sister groups that form the supraorder Amphiesmenoptera) produce silk fibers containing L- and H-fibroins in their labial glands and their major silk proteins have a common origin[12]. It has been suggested that the production of this type of silk has been conserved for over 300 million years[13].

The silk produced by lepidopteran caterpillars is secreted by a pair of specialized labial (salivary) glands called silk glands (SG). A mixture of silk proteins is stored in a lumen of the silk gland as a thick solution that solidifies after spinning. While the overall silk structure is similar, the individual proteins can vary greatly in Lepidoptera. Depending on solubility, silk proteins are traditionally divided into two groups: insoluble fibroin proteins, which form two core filaments and sericin proteins soluble in hot water which form a hydrophilic coating and seal the filaments into a fiber[14,15]. The fibroins are produced by the posterior part of the silk glands (PSG), while the coating proteins are progressively added to the stored silk in the middle silk glands (MSG). The anterior part of the silk glands (ASG) apparently does not produce any silk component and serves as the gland outlet[14]. The coating proteins show a great heterogeneity between species, both in the number of proteins they contain and in the sequences of the individual components[16].

Detailed analysis of silk components has been performed in several representatives of the moth families Bombycidae[17], Saturniidae[18], Pyralidae[19], Nolidae[5], Tineidae[7] and Psychidae[20], with most species belonging to the higher Ditrysia, which comprises more than 98% of the extant lepidopteran diversity[12]. However, the detection of silk proteins in newly studied species is hampered by the repetitive nature of their sequences and low sequence similarity even between species from the same family. Gene losses and duplications combined with rapid sequence changes make identification of individual genes encoding coat proteins based on similarities quite difficult[16]. Consequently, full-length sequences and exon-intron architecture of the silk genes are often missing.

The superfamily Yponomeutoidea belongs to one of the early diverged ditrysian lineages[21]. Members of the family Yponomeutidae are known from the Lower Cretaceous[22,23] which makes it one of the first groups to shift from internal to external feeding[24] colonizing herbs, shrubs, and trees on a large scale[25]. European ermine moths of the genus Yponomeuta have long been studied in terms of the chemical ecology of insect-plant associations and ecological speciation through host switching[26–29].

The genus Yponomeuta includes 76 described species distributed all around the world (except South America and Arctic regions)[30]. Small ermine moths are well recognized for their extensive protective webbing, produced by larvae of some species. During occasional outbreaks, the larvae can defoliate entire trees or bushes and cover them with silk within days. These webs consist of interwoven silk threads spun around the branches of food plants. Larvae pupate in the protective nests in cocoons that are slightly transparent and consist of a single layer of silk (Fig. S1)[31,32]. A previous study in the bird-cherry ermine moth, Yponomeuta evonymella, examined abundant silk gland-specific cDNAs and discovered transcripts for the light chain of fibroin (L-Fib), fibrohexamerin (Fhx or P25), and a partial sequence for the heavy chain of fibroin (H-Fib), suggesting that the composition of silk thread core is conserved[33]. However, other silk components were not identified.

In the present study, we show a detailed analysis of silk genes and their products in a closely related spindle ermine moth, Yponomeuta cagnagella, and extend the earlier analysis of Y. evonymella. To identify and characterize all genes encoding major silk components, we sequenced and assembled the genome of Y. cagnagella and combined the results with transcriptomic and proteomic analyses and a homology searches using proteins detected in Y. evonymella.

## Results

*Y. cagnagella* genome. The genome size of *Y. cagnagella* was determined by flow cytometry. Four heads of adult males were used with *Ephestia kuehniella* standards in independent replicates. The resulting genome size was $709.54 \pm 6.52$ Mbp (mean ± standard error, N = 4). An example of flow cytometry analysis is shown in Fig. S2. In addition, the genome size was estimated from k-mer spectra, in this case reaching 508.47 Mbp (Fig. S3). When referring to genome size below, we use the flow cytometry result as we consider it to be more accurate.

Overall, DNA sequencing yielded 26 Gbp Oxford Nanopore (ONT) long reads (N50 = 19.1 kb) and 43.8 Gb Illumina short reads, corresponding to approximately 37-fold and 62-fold genome coverage, respectively. After preprocessing of the raw data, the final dataset consisted of 25.5 Gb corrected long reads with N50 = 19.5 kb, corresponding to approximately 36-fold genome coverage.

The primary genome assembly generated with Flye resulted in 30,252 fragments comprising 978.3 Mb and an N50 of 67.3 kb. The completeness of the assembly was assessed by BUSCO using the Insecta Core Orthologue Database and revealed a high proportion of duplicated genes (44.4%). Together with the length of the assembly exceeding the genome size and the results of the k-mer analysis showing high heterozygosity (1.29%; Fig. S3), this suggested significant presence of haplotypic duplications.

After deduplication with the purge dups pipeline and further polishing with Illumina data, the final assembly of the *Y. cagnagella* genome consisted of 11,790 contigs comprising 626.3 Mb with contig N50 = 96.5 kb. The longest fragment was 931.2 kb in length. In the final assembly, 96.9% of BUSCO orthologs were complete, 1.6% were fragmented and only 1.5% were missing. The number of duplicated genes decreased to 10.5%, indicating that duplicated haplotigs caused by high heterozygosity were significantly reduced. Kraken 2 search for contaminants against the standard library identified 9 contigs that were assigned to humans and 23 that were assigned to bacteria (Table S2). However, detailed manual inspection did not confirm any contamination as majority of k-mers of these contigs remained unclassified, suggesting these sequences may be derived from repeats. Furthermore, blastn search of the sequences in the nt database revealed lepidopteran sequences as the closest matches in all cases.

Annotating the final *Y. cagnagella* assembly showed that 46.75% of the sequence was masked as repetitive elements, with genome GC content of 39.21%. The most abundant repeat class was long interspersed nuclear elements (LINEs; 10.69% of the genome), followed by short interspersed nuclear elements (SINEs; 5.06%) and DNA elements (3.33%). Only 0.57% of the assembly

was classified as satellite DNA (Table S3 and S4). The soft-masked assembly was deposited in the Dryad repository (https://doi.org/10.5061/dryad.4j0zpc8d3). In total, 30,003 protein-coding gene models were predicted by the BRAKER pipeline, with average gene length of 8,331 bp and on average 5.9 exons per gene model. BUSCO assessment of completeness showed 93.5% of Insecta orthologs to be present in the gene set.

Finally, we compared the ermine moth assembly to other lepidopteran genomes (Table 1). *Y. cagnagella* represents one of the largest and most heterozygous genomes from the list. The difference in genome size and assembly length in *Y. cagnagella* is close to *S. exigua* (88.3% and 88.2%, respectively) and the N50 statistic is similar to those obtained for genome assemblies of other non-model lepidopterans, namely *D. arcuata*, *E. variegata* and *M. jurtina*, sequenced by cost-effective low coverage sequencing strategy without any further scaffolding techniques such as HiC. As only 1.5% insect BUSCO orthologs were missing, the *Y. cangnagella* genome assembly represents a valuable resource for further studies.

***Y. cagnagella* silk glands and silk**. The silk glands of *Y. cagnagella* show characteristic morphology with distinct parts of the ASG, MSG and PSG. Figure 1a shows three compartments of silk glands with a thin anterior part, the thickest middle part with a wide lumen and a slightly thinner posterior part consisting of large secretory cells surrounding the thin lumen. To study the morphology of SG in relation to its position in the body, we made a series of transverse sections through the larvae in the last-instar. The transverse sections revealed further morphological differences in the different parts of the glands as well as the structure of the lumen with a central localization of the fibroin components covered by envelope layers composed mainly of sericins (Fig. 1b–d). The ratio between the thickness of the fibroin and sericin layers varies along the length of SG. The sericin layer is not present in the PSG (Fig. 1d, e). The silk in the lumen of the gland shows color variations between the different glandular compartments when stained with Masson trichrome stain. The color reflects changes in the structure or pH of the silk components.

Scanning electron microscopy was used to examine the external and internal structure of the cocoon. The silk filaments are relatively thin (2.6–2.8 μm) and form doublets as a product of paired glands. Unlike *B. mori* or *G. mellonella*, in which the silk filaments form a dense network with a large amount of glue material covering the inner surface of the cocoon, the silk of *Y. cagnagella* has a rather grid-like character, occasionally connected by a tuft of adhesive (Fig. 2a–d). The relatively low adhesive content is also evident in cross sections stained with toluidine blue and examined by light microscopy (Fig. 2e, f). Based on the different staining intensity, one can distinguish between the dark fibroin core and the light sericin envelope. For comparison, Fig. 2f also shows a section through a cocoon of the waxmoth (*G. mellonella*), which contains a high proportion of sericins. Also, *Y. cagnagella* silk fibers are strikingly thinner compared to *G. mellonella* although the larvae have rather similar size (Fig. 2e, f).

**Transcriptome assembly and proteomic analysis of cocoon proteins**. To identify candidate genes encoding silk proteins, we prepared a silk gland-specific transcriptome. A total of 7.85 million paired-end reads were obtained and used to assemble the transcriptome after cleaning. The trancriptome assembly was deposited in the Dryad repository (https://doi.org/10.5061/dryad.4j0zpc8d3). A total of 1155 (84.4%) of 1367 BUSCOs were identified in the assembly. 1045 (76.4%) of these were present as single copies, while 110 (8.0%) existed as multiple copies in the assembly. 66 BUSCOs (4.8%) were found in fragmented form, while 146 BUSCOs were missing (10.8%). We concluded that the 146 missing BUSCOs are likely the result of high SG specialization for silk production. 26,054 open reading frames (ORFs) were found in the assembly. Within these ORFs, 987 signal peptides were identified. Several highly abundant transcripts showed homology to the known silk proteins from *B. mori*, including fibroin heavy chain H-Fib, L-Fib, Fhx and seroins 1, 2 and 3.

To identify the entire set of silk components, we analyzed the cocoon proteins by mass spectrometry (MS). The silk proteins were identified by tryptic peptide mapping. We detected more than 500 peptides that were mapped to 120 proteins. Some of the

**Table 1 Assembly metrics of *Y. cagnagella* genome and selected lepidopteran genome assemblies.**

| Species (Family) | Genome size (Mbp) | Assembly length (Mbp) | Num. of fragments | N50 (Mbp) | Heter. (%) | Repeats (%) | BUSCO (Insecta) | Predicted genes |
|---|---|---|---|---|---|---|---|---|
| **SPECIES OF INTEREST** | | | | | | | | |
| ***Yponomeuta cagnagella*** (Yponomeutidae) | 709.5 | 626.3 | 11,790 | 0.1 | 1.3 | 46.7 | C: 96.9% [S: 86.4%, D: 10.5%] F: 1.6% M: 1.5%[a] | 30,003 |
| **MODEL SPECIES** | | | | | | | | |
| ***Bombyx mori*** (Bombycidae) | 508.6[b] | 460.3 | 696 | 16.8 | NA | 46.8 | C: 99.0% [S: 97.9%, D: 1.1] F: 0.3%, M: 0.7%[a] | 16,880 |
| ***Plutella xylostella*** (Plutellidae) | 339.4 | 328.8 | 443 | 11.1 | 0.01 -0.04 | NA | C: 99.2% [S: 98.0%, D:1.2%] F: 0.3%, M: 0.5%[a] | 19,002 |
| **ONT-BASED ASSEMBLIES** | | | | | | | | |
| ***Drepana arcuata*** (Drepanidae) | 303.2 | 270.5 | 11,493 | 0.1 | 1.0 | 8.3 | C: 86.5 % [S: 84.3%, D: 2.2%] F: 4.2, M: 9.3%[a] | 10,398 |
| ***Ephestia elutella*** (Pyralidae) | NA | 576.9 | 804 | 19.0 | NA | 58.1 | C: 92.9% [S: 91.9%, D: 1%] F: 2.2%, M: 4.9% | 15,637 |
| ***Eumeta variegate*** (Psychidae) | 708.7 | 731.9 | 12,720 | 0.3 | 0.9 | NA | C: 94.9% [S: 91.5%, D: 3.4%] F: 1.4 %, M: 3.7%[a] | 36,444 |
| ***Manduca sexta*** (Sphingidae) | 420.5[b] | 470.0 | 4,057 | 14.3 | NA | 33.9 | C: 98.1% [S: 93.1%, D: 5%] F: 0.7%, M: 1.2% | 25,256 |
| ***Maniola jurtina*** (Nymphalidae) | 576.4 | 618.4 | 20,143 | 0.2 | 1.9 | 75.6 | C: 88.7% [S: 68.9%, D: 19.8%] F: 3.5%, M: 7.7% | 38,101 |
| ***Sesamia nonagrioides*** (Noctuidae) | 943.5 | 1,021.3 | 2,553 | 1.1 | NA | NA | C: 98.5% [S: 94.1%, D:4.4%] F: 0.7%, M: 0.8%[a] | 17,230 |
| ***Spodoptera exigua*** (Noctuidae) | 370.0 | 419.3 | 946 | 1.1 | NA | NA | C: 99.4% [S: 96.5%, D: 2.9%.] F: 0.2%, M: 0.4%[a] | 18,477 |

*Heter.* heterozygosity, *NA* not available.
[a]BUSCO run in this study (see Materials and Methods).
[b]Animal Genome Size Database (https://www.genomesize.com/).

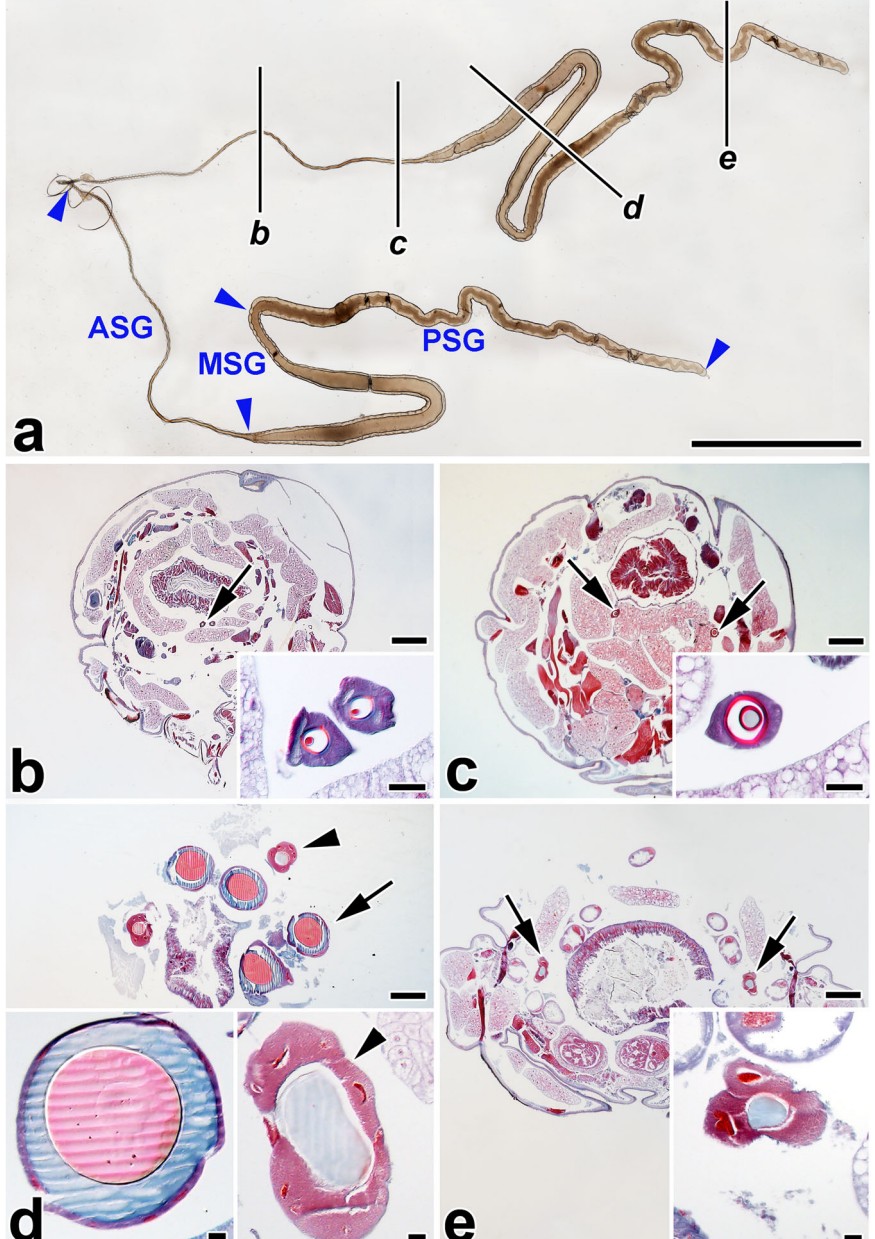

**Fig. 1 Morphology of the silk gland (SG) from *Yponomeuta cagnagella* last instar larvae. a** Overview of whole mounted SG. Blue arrowheads show the boundaries of the SG compartments, where ASG—anterior SG, MSG—middle SG and PSG—posterior silk gland part. Black lines marked by small italicized letters *b–e* refer to the whole-body sections, and show the approximate position where the glands were cut in respective transverse paraplast sections. **b–e** Transverse paraplast sections through the body of 5th larval instar stained with Masson trichrome stain (Sigma-Aldrich). Inserted images show higher magnification of the SG section pointed by the arrows and arrowheads. **b, c** Section through the body and anterior silk gland. **d** Middle silk gland. **e** Posterior silk gland. Arrowheads depict the section of the posterior silk gland, arrows in **b** and **c** show ASG and arrows in **d** depict the section of the posterior part of the middle silk gland. Red and blue areas differ in structure. Scale-bars: (**a**) 1000 μm; (**b–e**) 200 μm; inset images, 20 μm.

detected proteins lacked the signal peptide and they mainly represented enzymes and housekeeping proteins. A total of the 79 discovered proteins contained putative signal peptides. Some of them had close homologs in other species that were not associated with silk and were excluded from further analysis. Because we assume that the structural proteins of silk are highly abundant, we excluded several candidates that were under-represented. The remaining 30 candidate transcripts were used for further analysis.

To identify the silk proteins of *Y. evonymella*, we performed a proteomic analysis of the cocoon silk as described for *Y. cagnagella* and discovered 130 proteins, 39 of which contained

signal peptides. Comparison with the *Y. cagnagella* data revealed a very similar set of proteins to the previous analysis and three additional proteins. These proteins were not detected in the proteomic analysis in *Y. cagnagella*. However, homologous sequences encoding these proteins were present in the spindle ermine moth transcriptomic database and are listed in Table 2.

**Transcription specificity of silk candidate proteins and similarity between *Y. cagnagella* and *Y. evonymella* candidate silk genes.** The putative secretory proteins detected in cocoon silk could still contain nonspecific components produced outside SG

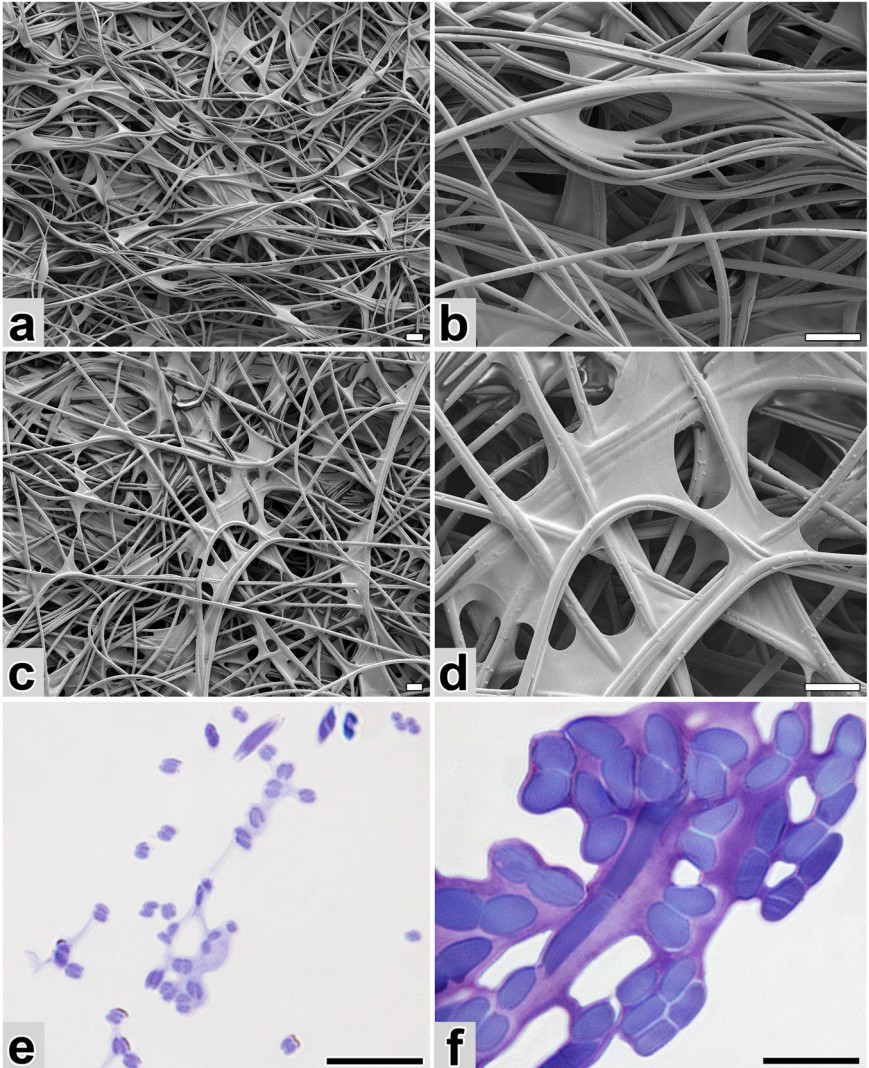

**Fig. 2 *Y. cagnagella* and *G. mellonella* cocoons. a, b** Scanning electron microscopy of different magnification of the outer side of the *Y. cagnagella* cocoon. **c, d** Images of the inner side of the *Y. cagnagella* cocoon. **e, f** Comparison of silks from *Y. cagnagella* and *G. mellonella*. Semithin cocoon sections of *Y. cagnagella* (**e**) and *G. mellonella* (**f**) stained with toluidine blue. Note the different fiber thickness and the content of glue-like material (light purple). Scalebars: 10 μm.

(e.g. material deposited from the digestive tract) or proteins produced at higher levels in various other tissues. Because tissue-specific expression of the homologous proteins of H-Fib and Fhx (P25) has already been studied in detail in other moth species, we focused on other candidates. First, we examined 11 candidate gene products by northern blotting.

To compare the size of the homologous transcripts and to verify the evolutionary conservation of the studied genes, we also included RNA from the related *Y. evonymella* as a control. We prepared labeled probes of *Y. cagnagella* cDNAs using RT-PCR with the primers listed in Table S1. The signals in *Y. cagnagella* were almost as strong as those from *Y. evonymella* indicating high similarity of the sequences examined between the two species (Fig. 3). The size of Ser1 and Ser2 is similar in both species, whereas Muc1 and Ser4 were smaller in *Y. evonymella*. The Ser3 signal was weaker in *Y. cagnagella* suggesting lower expression. In contrast, fibrillin1-like (FB1L) showed lower intensity in *Y. evonymella.*, which might be related to lower expression level or sequence diversification between *Y. cagnagella* and *Y. evonymella*.

In addition to northern blots, we compared the expression of a total of 17 candidate genes in SGs, intestines, fat bodies, and integuments by real-time PCR (Fig. 4). Northern analysis and qPCR results confirmed SG-specificity of 21 genes, including Fhx (P-25), small SG peptide 1, and genes encoding various enzymes such as proteinase, sulfatase, and peroxidase. In contrast, antichymotrypsin1 (ACHP1), trypsin-like a (TRPLa), fibrillin2-like (FB2L), bangles and beads (BNB), and agrin-like (AGRL) were not specific to silk glands. Our final list also includes the known silk components H-Fib and seroins, making the final number of silk components 25 (Table 2).

Finally, we estimated the evolutionary divergence and similarity between DNA and amino acid sequences by comparing the homologs of eight proteins and their cDNAs detected in the silks of *Y. cagnagella* and *Y. evonymella*. A summary is provided in Table S5 and shows 96-99% identity between the sequences of the two species. This supports the idea that individual silk components can be easily identified between the two species from the same genus based on similarities. Interestingly, all four sequences encoding structural silk components tended to have a

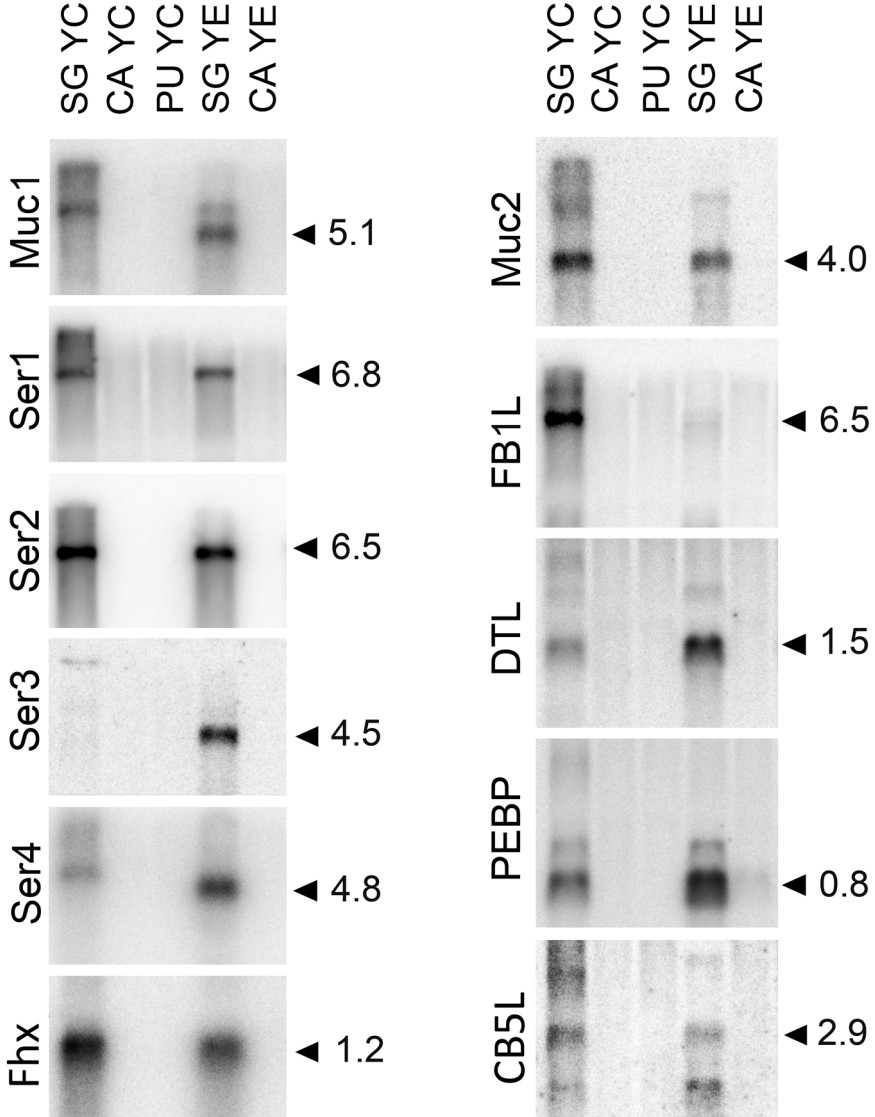

**Fig. 3 Tissue specificity analysis of mRNA expression of new silk gene candidates by Northern blotting.** Lanes: SG YC—silk glands of *Y. cagnagella* spinning last instar larva; CA YC—carcass (ablated SG) of *Y. cagnagella*, PU YC—pupa of *Y. cagnagella*, SG YE—silk glands of *Y. evonymella* spinning last instar larva; CA YE—*Y. evonymella* carcas without SG. Total RNA (5 µg) was separated on an agarose gel using electrophoresis, blotted to a nylon membrane and probed with [$^{32}$P]-labeled cDNA fragments from indicated *Y. cagnagella* genes. The size of the detected transcripts is indicated. Gene names are shown in Table 2.

greater genetic distance from their counterparts from *Tineola bisselliella* than four metabolic genes used as controls (Table S5).

**Obtaining complete sequences and elucidating the exon-intron structure of genes encoding silk components.** The resulting number of transcripts encoding major silk components in *Y. cagnagella* appears to be similar to that of *G. mellonella* or *T. bisselliella*[7,16]. To learn more about the structure of these genes and their relationship to the silk genes of other moth species, we determined their exon-intron organization and complete sequences, including UTRs, by placing the CDS on the genomic sequence.

As shown in Table 2, we have identified the complete sequences of 25 genes encoding the most abundant *Y. cagnagella* silk components, including fibroin-heavy chain, fibroin-light chain, P25/Fhx, four or five putative sericins, three seroins, four zonadhesin-like proteins, two mucins and several other proteins, including sulfatase (SLP), peroxidase (PXD), serine protease-like (SPL), dentin-like protein (DTL), cytochrome b5-like protein

(CB5L), phosphatidylethanolamine-binding protein homolog (PEBP) and fibrillin1-like (FB1L). Their annotated sequences were deposited in GenBank and schematic drawings of the structures of each gene are shown in Fig. S4. Three additional silk-associated sequences of *Y. cagnagella* were detected based on homology to *Y. evonymella*, including venom serine protease 34-like, trypsin-like proteinase T2a, antichymotrypsin-1-like b and antichymotrypsin-2-like. Two of them may not be silk gland-specific, as antichymotrypsin 2-like (ACHP2) and trypsin-like b (TRPLb) are very close orthologs of the *Y. cagnagella* genes, which were examined above by PCR as not being silk specific.

*H-Fib* encodes the largest silk protein of 486 kDa, characterized by high glycine and alanine content (29.4 and 26%, respectively). The gene consists of two exons and one intron. The first exon is very short and encodes 10 N-terminal amino acid residues of the signal peptide. The second exon is very large and its ORF encodes 5753 amino acid residues. The intron is 340 bp long. The central part of H-Fib molecule consists of an

**Table 2 List of *Y. cagnagella* silk gene candidates, their GenBank accession codes and their putative products with conspicuous expression in silk glands.**

| Gene name | Gene symb. | GenBank# | Exons | Prot Mw | Numb. a. a. | Major a.a. | Hydropath. | pI |
|---|---|---|---|---|---|---|---|---|
| Fibroin H | YC-FibH | MZ959145 | 2 | 486140.9 | 5763 | Gly (29.4) | 0.025 | 4.00 |
| Fibroin L | YC-FibL | MZ959144 | 7 | 26094.12 | 258 | Ala (20.2%) | 0.205 | 4.34 |
| Fibrohexamerin | YC-Fhx | MZ959146 | 5 | 24953.41 | 217 | Leu (10%) | −0.065 | 5.16 |
| Sericin 1 | YC-Ser1 | MZ927540 | 8 | 105259.4 | 1061 | Gln (28.9) | −1.725 | 6.46 |
| Sericin 2 | YC-Ser2 | OK019717 | 3 | 163190.9 | 1595 | Ser (43.7%) | −1.761 | 6.13 |
| Sericin 3 | YC-Ser3 | OK019718 | 3 | 87831 | 976 | Ser (62.4%) | −1.087 | 3.78 |
| Sericin 4 | YC-Ser4 | OK019719 | 8 | 80018.09 | 731 | Asn (28.9%) | −2.133 | 4.23 |
| Mucin1 | YC-Muc1 | MZ981775 | 32 | 358281 | 3262 | Ser (22%) | −1.414 | 4.36 |
| Mucin2 | YC-Muc2 | MZ981776 | 25 | 110178 | 985 | Gln (17.1) | −1.271 | 5.37 |
| CytB-like | YC-CB5L | MZ981777 | 5 | 18466 | 167 | Leu (9.6) | −0.325 | 4.88 |
| Zonadhesin 1 | YC-ZNL1 | MZ959140 | 3 | 17537,14 | 157 | Cys (12.7%) | −0.398 | 8.03 |
| Zonadhesin 2 | YC-ZNL2 | MZ959141 | 6 | 41007,39 | 372 | Cys (13.4) | −0.548 | 5.49 |
| Zonadhesin 3 | YC-ZNL3 | MZ959142 | 20 | 137569,09 | 1253 | Cys (15.2) | −0.663 | 5.33 |
| Zonadhesin 4 | YC-ZNL4 | MZ959143 | 8 | 46321,88 | 426 | Cys (14.1) | −0.666 | 4.99 |
| Seroin1 | YC-Srn1 | MZ959147 | 10 | 23259 | 218 | Pro (12.4%) | −0.501 | 6.42 |
| Seroin2 | YC-Srn2 | MZ959148 | 3 | 16855.55 | 156 | Gln (15.4%) | −0.45 | 5.88 |
| Seroin3 | YC-Srn3 | MZ959149 | 4 | 11203.71 | 103 | Asp (11.7%) | −0.385 | 5.14 |
| Dentin-like | YC-DTL | MZ981778 | 3 | 51554.99 | 461 | Glu (16.3%) | −1.529 | 4.31 |
| PEBP-binding p. | YC-PEBP | MZ981779 | 4 | 21058 | 187 | Leu (10.2%) | −0.088 | 6.28 |
| Serine protease | YC-SPL | MZ981780 | 4 | 17070.59 | 152 | Leu (9.9%) | −0.114 | 5.74 |
| Venom alergen | YC-VAL | MZ981781 | 6 | 25478.52 | 219 | Ala (8.2%) | −0.553 | 6.70 |
| Sulphatase | YC-SLP | MZ981782 | 8 | 66182.86 | 595 | Leu (11.1%) | −0.207 | 5.16 |
| Peroxidase | YC-PXD | MZ981783 | 12 | 71135.39 | 623 | Leu (8.2%) | −0.49 | 5.31 |
| Fibrillin-like 1 | YC-FB1L | OK019714 | 14 | 127726.4 | 1193 | Ser (25.7%) | −1.471 | 5.08 |
| Small SG pep1 | YC-Ssp1 | OK019716 | 6 | 6445.35 | 58 | Gln (20.7%) | −0.04 | 5.82 |
| Venom prot | YC-VenP | OP251046 | 5 | 18928 | 169 | Leu (11.2%) | −0.28 | 7.63 |
| Trypsin l. | YC-TrpL | OP251047 | 10 | 43131 | 397 | Gly (9.6%) | −0.04 | 5.25 |
| Antichym2 | YC-Ach2L | OP251049 | 10 | 43451 | 386 | Leu (9.6) | −0.275 | 5.41 |

The number of exons was inferred from comparison of genomic and cDNA sequences. Protein mW, number of amino acids, major amino acid indicates the percentage of the most abundant residue; predicted molecular weight, hydropathicity and pI were determined using the ExPASy ProtParam tool (http://us.expasy.org/tools/protparam.html). The last three silk candidates (venom serine protease 34-like, trypsin-like proteinase T2a and antichymotrypsin-2-like) were not tested for SG tissue specificity.

imperfect repetitive sequence encoding 27-31 aa long repeat motifs. These motifs contain an SSAAA sequence reminiscent of the fibroin repeats of *G. mellonella* or *Antheraea yamamai* that form the crystalline regions responsible for fiber strength (Fig. S5).

There are at least four major sericin genes that encode large hydrophilic proteins localized on the surface of the silk fiber. They are characterized by a high content of serine residues and repetitive sequences. Sericin2 (Ser2) and sericin3 (Ser3) contain the highest proportion of serine residues and share a common organizational structure with three exons and two introns. Furthermore, the gene for Ser2 is located near Ser3 suggesting that they are derived from a single progenitor gene. In addition, sericin 1 contains a sequence encoding a characteristic CVCY motif located 17 amino acid residues away from the C-terminus. A similar motif is also found in the *B. mori*, *A. yamamai* or *G. mellonella* sericin 1 homologs (Fig. S6)[16,34].

We also found several other candidate structural silk components containing imperfect repetitive motifs, including two putative homologs of mucins, four zonadhesin-like proteins, a fibrillin 1-like (FB1L), and a dentin-like protein (DTL), see Table 2. Mucin1 (Muc1) is the second largest silk protein and contains 22% serine residues. The genes encoding Muc1 and mucin2 (Muc2) appear to contain a large number of exons (32 and 24, respectively; Fig. S4).

We found three seroin genes arranged in a tight cluster localized in a single genomic contig (contig_36475). We also detected *B. mori* homologs for venom allergen-like (VAL) and several enzymes, whose roles in silk are unclear. Finally, we confirmed the tissue specificity of the expression of short SG peptide 1 (SSP1) with unknown function, which contains a chain of glutamine residues (Table 2).

## Discussion

For the purpose of comprehensive silk analysis, we sequenced and assembled the draft genome and transcriptome of the spindle ermine moth, *Y. cagnagella*, a representative of early diverging family of ditrysian Lepidoptera. Using a cost-effective approach combining low coverage Oxford Nanopore and Illumina reads, we obtained a draft genome sequence, which allowed us to identify the full-length exon-intron architecture of 25 genes encoding silk in this species.

Genome size measured by flow cytometry (710 Mb) differed from the k-mer survey estimate (508 Mb), but the latter approach may have been affected by data quality or sequencing depth[35]. Thus, we assume the flow cytometry results to be more accurate. Yet, both measurements suggest that genome size of *Y. cagnagella* is larger than average (1C = 430 Mb) or the ancestral size (1C = 489 Mb) proposed for Lepidoptera[36]. This high value can be explained by expansion of repeats, as nearly half of the assembly was annotated as repetitive elements, and/or by large intron size as suggested by Chen et al.[37] in the webworm *Hyphantria cunea*. The primary genome assembly length was 978.3 Mb, which was significantly larger than expected genome size and probably the result of haplotype retention caused by high levels of heterozygosity revealed in the sample by k-mer analysis. After deduplication, the final 626.3 Mb assembly length was nearly 90% of the expected genome size and contained 96.9% BUSCO orthologs. Approximately 30,000 protein-coding gene models were predicted in the assembly, comparable to the range found in other lepidopteran assemblies. The average gene length was 8.3 kB, as expected for the *Y. cagnagella* genome size[38]. The average number of 5.9 exons per gene model was similar to 6.0 and 6.1 exons in *Plutella xylostella*[39] and *Bombyx mori*[40], respectively. 93.5% BUSCO orthologs were present in the final gene set. Overall, the quality of obtained ermine moth genome is

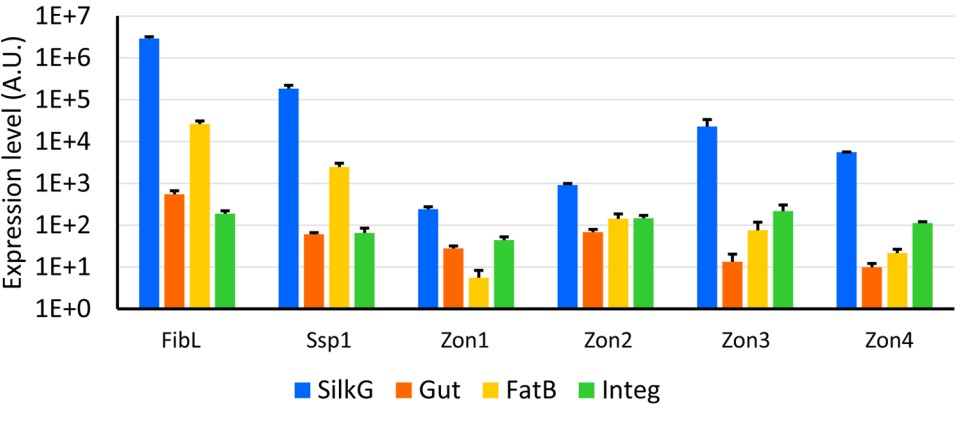

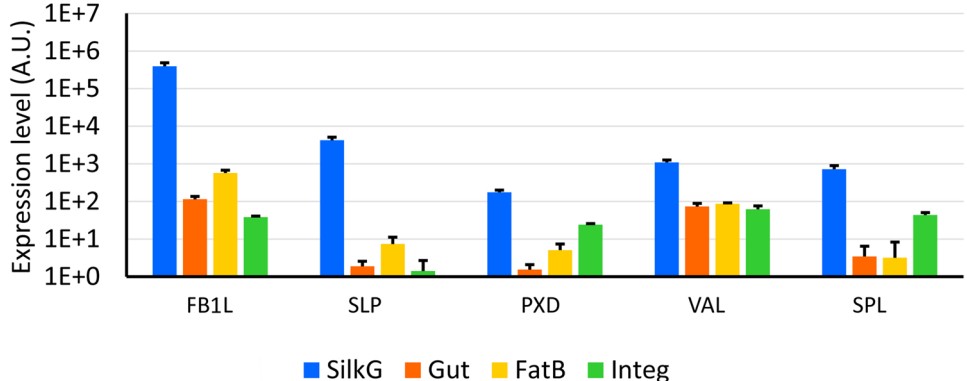

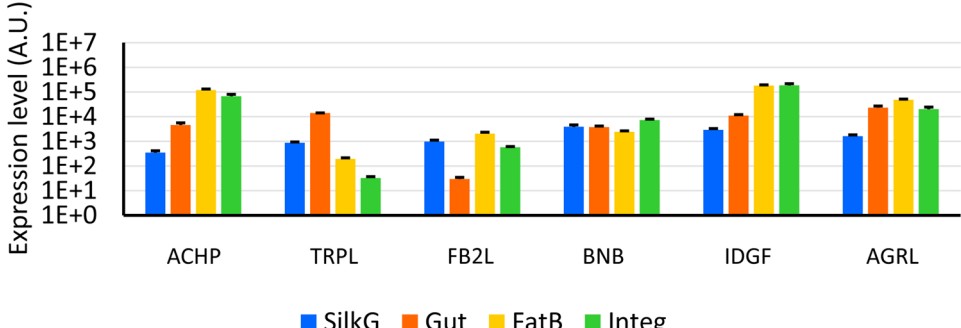

**Fig. 4 Tissue specificity of expression of 17 selected genes in four tissues.** (silk glands—blue, gut—orange, fat body—yellow and integument—green) using qPCR ($n = 3$, mean ± s.d.). The mRNA expression levels of these genes were normalized with an external control gene (Elongation factor 1-α) and calculated with 2−ΔCt relative quantification. Full gene names are shown in Table 2.

comparable to other non-model Lepidoptera assemblies without higher scaffolding[41,42] and represents a valuable resource for genomic analyses, even for studies of large genes with repetitive domains such as fibroins, which often present a great challenge.

As mentioned above, silk contains two core filaments covered by a hydrophilic coating, which seals the filaments into a fiber. It has been shown in previous studies that silk core contains H-Fib, L-Fib and Fhx produced and assembled in PSG[43–45]. The three-component complex of silk fibroins is conserved among most lepidoptera[46–48].

H-fib is the major structural module of the fibroin core. It is encoded by a large single-copy gene containing conserved sequences at both ends and a region of repeated sequences in the central part of the molecule that are highly species-specific and encode structural components called crystalline blocks separated by amorphous regions (Fig. S5). These blocks are generally

responsible for the tensile strength of silk fibers and mostly consist of the amino acid residues glycine, alanine, and serine[49]. Adjacent chains of crystal domains are held together by strong hydrogen bonds in an antiparallel arrangement, forming β-sheets[50]. Different lepidopteran silks can be classified into three main categories based on the distance between β-sheets, which depend on the arrangement of the three major amino acid residues. Class I repeats consist of alternating amino acids Gly and Ala, class II consist of ((Gly-Ala)x -Ala)n, and class III repeats are composed predominantly of Ala or Ala and/or Ala-Ser chains[46–49,51] (Fig. S5). The central region of the H-fib of *Y. cagnagella* contains SSAAA sequence motifs similar to those of the H-fib of *A. yamamai* or *G. mellonella*, and the H-fib of all three species belong to the X-ray class III[49,52] (Fig. S5).

Conserved sequences at both ends of H-Fib (Fig. S7 and S8) are likely involved in interactions with L-Fib and Fhx proteins[53]. It

was shown that *B. mori* ortholog of L-Fib binds to the C-terminus of H-Fib by disulfide bond and this complex is required for fibroin transport from the endoplasmic reticulum[54]. Both L-Fib and Fhx do not contain repetitive sequences (Fig. S9 and S10). L-Fib has been reported to be present in silks of almost all Lepidoptera (except for moths of the family Saturniidae in which it was lost) as well as caddisfly species[46,55]. *Y. cagnagella* L-Fib is encoded by a single-copy gene and the protein shows 47% identity with L-Fib from *T. bisselliella* and 39% identity with *B. mori* (alignment of several L-Fib protein sequences from different lepidopteran species and one caddisfly is shown in Fig. S9).

*Y. cagnagella* Fhx shows 62% identity with Fhx from *T. bisselliella* and 39% with *B. mori* (alignment of several *Fhx* genes from different lepidopteran species is shown in Fig. S10). The *B. mori* ortholog of Fhx was reported earlier to bind noncovalently to the N-terminal part of H-Fib and is involved in its transport from the endoplasmic reticulum and is also involved in maintaining the solubility of secretory fibroin granules in the lumen of SG[56]. In *Pseudoips prasinana* there are 6 paralogs of *Fhx* gene, while *Fhx* orthologs seem to be missing in *P. californicus* and in members of Saturniidae family[33,46]. In addition to the true Fhx proteins, most moths also possess distantly related Fhx-like proteins of unknown function that form a separate subfamily[5], e.g. two of them in *T. bisselliella*, eight in *P. prasinana*, and six in *B. mori*[5]. Interestingly, we have not found a homolog of such *Fhx*-like gene in *Y. cagnagella*.

Silk-coating proteins, which are produced in MSG, are represented mainly by sericins and mucins. They undergo even more profound evolutionary changes than the fibroin subunits, including frequent gene duplications and losses. The genomic structures of *Y. cagnagella* mucins and sericins suggest that they use different strategies to encode repetitive regions. Fig. S6 shows an example of the enormous divergence of the putative Ser1 protein, which can be recognized by CXCY motif close to its C-terminus (Fig. S6). Similarly, members of mucin-1-like family contain three-cysteine motif CXCYCZ (Fig. S11). Interestingly, while sericin repeats are arranged in tandems in large exons, mucin 1 tends to duplicate entire exons (Fig. S4). The alignments of sericin-1 and mucin-1 like proteins shown in Fig. S6 and S11 reveal consensus sequences, which may allow detection of orthologs in other ditrysian Lepidoptera. Accurate alignment of other sericin and mucin repeats for evolutionary studies is very difficult and will require more data. The species-specific branching of sericin proteins (especially those containing high percentage of serine residues) in previously reported cladograms[9] suggests that there were multiple independent duplication events in the evolution of these genes. Such duplications were suggested previously for sericins in *G. mellonella, A. yamamai* or *Samia cynthia ricini*[16].

Several proteins, including mucins and zonadhesin-like sequences detected in the silk coating of *Y. cagnagella* have homologs in the silks of other moth species and thus appear to be regular components of Lepidoptera silk. Both mucins and zonadhesin-like proteins could have an adhesive role(s) or be involved in antimicrobial protection. Zonadhesin-like proteins are characterized by Til and EGF domains and by highly repetitive sequences with cysteine as the most abundant amino acid residue, accounting for 12–14% of the total amino acid residues (Table 2). The genes encoding zonadhesin-like proteins vary in size, so that the larger ones (Zon3) appear to be the duplicated products of shorter precursor forms (similar to Zon1) (Fig. S4). Mucins have also been found in labial glands of other insects and contain tandemly repeated sequences ProThrSer[57].

The role of less abundant proteins annotated as fibrillin-1-like, dentin-like or venom allergen-like proteins in the silk structure of *Y. cagnagella* is unclear, and their putative homologs in other species, including *B. mori* have not previously been associated with silk. Their silk gland specificity, hydrophilicity and repetitive sequences link them to other structural coating proteins. The function of other components, including cytochrome b5 (Cyt b5), sulfatase (SLP) and peroxidase (PXD), which contain signal peptides and are present in silk of some other moths, remains to be elucidated. We need data from additional lepidopteran species to determine whether they represent a primitive trait, a specific adaptation of the superfamily Yponomeutoidea, or regular silk components.

Not all proteins detected in cocoons are structural silk components produced by SG, silk may also contain proteins from the digestive tract or housekeeping proteins from SG cells that enter the lumen by apocrine-like secretion, similar to the salivary glands of *Drosophila melanogaster*[58]. Further analysis of candidate gene products for SG-specific expression by qPCR and/or northern blotting is required. In this study, we eliminated 6 candidate proteins from 31 secretory proteins detected by proteomics as nonspecific for silk glands. In a similar approach in *T. bisselliella*, as much as 50% of the secretory proteins detected in the cocoon were found to be nonspecific[7].

More information on silks produced by representatives of major evolutionary Lepidoptera clades will allow us to improve our knowledge of silk structure, our search for new biomaterials, and more accurate gene annotations in future sequencing projects. Overall, we sequenced and assembled the genome of *Y. cagnagella* and combined the results with transcriptomic and proteomic analyses to identify all major genes encoding silk components. In addition to the omics analyses we incorporated information on the silk of a related species, *Y. evonymella* analyzed simultaneously, which allowed us to complement the results by homology searches. This resulted in the identification of three additional genes. We provide a detailed annotation of 25 major silk structural genes, including their complete sequences and exon-intron structures. Omics methods allow detailed comparisons of the silks of different moths to search for the evolutionary origins and functional adaptations of individual silk components. This study fills an important gap in our growing understanding of silk gene structure, evolution and function and lays the foundation for future detailed comparative studies.

## Materials and Methods

**Insect material**. For flow cytometry, we used *Yponomeuta cagnagella* (Hübner, 1813) males from laboratory mass rearing started with larvae collected in Levín (Czech Republic). For sequencing and proteomic analyses, *Y. cagnagella* egg batches were collected in Watergraafsmeer (Amsterdam, The Netherlands). Hatched larvae were reared on twigs of their food plant, *Euonymus europaeus* (Linnaeus, 1753), until pupation. Pupae were sexed by their morphology, frozen in liquid nitrogen and stored for DNA extraction at −80 °C. *Yponomeuta evonymella* (Linnaeus, 1758) larvae were collected in Amsterdam (The Netherlands) and Vrabce (Czech Republic). Adults of the Mediterranean flour moth *Ephestia kuehniella* (Zeller, 1879; Lepidoptera, Pyralidae) were obtained from laboratory wild-type strain WT-C[59]. Larvae of the waxmoth *Galleria mellonella* (Linnaeus, 1758; Lepidoptera, Pyralidae) were from laboratory strain that was originally established from specimens found in Ceske Budejovice (Czech Republic).

**Histology and electron microscopy**. Wholemount preparation of the SG: The SGs were dissected and transferred to a drop of phosphate buffered solution on a microscopy slide, covered with a cover slip and imaged under Olympus BX63 microscope (Olympus, Hamburg, Germany) equipped with CCD camera (Olympus DP74). The final photograph was reconstructed by stitching of set of frames that represent projection of several Z-stack images using CellSens software (Olympus).

Paraplast sectioning of the 5th instar larvae: The cuticle of water anesthetized larvae were pierced under the fixative based on saturated picric acid, 3.6% formaldehyde and 2.3% of copper acetate supplemented with mercuric chloride (Bouin-Hollande solution)[60]. After one hour fixation the larvae were cut into three parts and subsequently fixed overnight at 4 °C. Standard techniques were used for tissue dehydration, embedding in paraplast, sectioning at 7–10 μm, deparaffinization, and rehydration. The sections were treated with Lugol's iodine followed by 7.5% solution of sodium thiosulphate to remove residual heavy metal ions, and then washed in distilled water. Staining was performed with HT15 Trichrome Stain (Masson) Kit (Sigma-Aldrich, Burlington, USA) according to the

manufacturer's protocol. Stained sections were dehydrated, mounted in DPX mounting medium (Fluka, Buchs, Switzerland). High-resolution images were captured by stitching several frames using a BX63 microscope, a DP74 CMOS camera, and cellSens software (Olympus).

Semithin sections of cocoons: Parts of cocoons embedded in resin (Epon) were cut with a glass knife and stained with toluidine blue. Samples were viewed and imaged under the BX51 microscope BX51 (Olympus, Hamburg, Germany) equipped with the DP74 CMOS camera (Olympus, Hamburg, Germany).

Silk ultrastructure: Pieces of cocoon were glued to aluminum holders, sputter-coated with gold, and analyzed using a Jeol JSM-7401F scanning electron microscope (Jeol, Akishima, Japan).

**Flow cytometry.** *Y. cagnagella* male genome size was determined from brain tissue by flow cytometry following[61] using *E. kuehniella* males as an internal standard (1C = 440 Mbp;[62]). Briefly, a fresh head of *Y. cagnagella* male was chopped along with a head of *E. kuehniella* standard using a razor blade in 500 µL of nuclei isolation buffer (0.1 M Tris-HCl pH 7.5, 2 mM $MgCl_2$, 1% Triton X-100;[62]. The suspension was filtered and 500 µL of nuclei isolation buffer was added. Samples were stained with propidium iodide (50 µg/mL) for 20 min and analyzed with a Sysmex CyFlow Space flow cytometer (Sysmex Partec, Münster, Germany) equipped with a 100 mW 532 nm (green) solid-state laser. Fluorescence intensity and side scattered light (SSC) of at least 8,000 nuclei was recorded and analyzed using FlowJo 10 software (TreeStar, Inc., Ashland, OR, USA). Mean, coefficient of variation, and number of analyzed nuclei were recorded for 2C peaks of both the sample and the standard, and the standard/sample ratio of mean fluorescence was calculated.

**Extraction of genomic DNA and tissue-specific RNA.** Genomic DNA for genome assembly and Illumina sequencing was extracted from a single male pupa by CTAB extraction[63]. High molecular weight DNA for Nanopore sequencing was extracted from three male pupae using the MagAttract HMW DNA kit (Qiagen, Hilden, Germany) according to the manufacturer's instructions.

For analysis of silk-related genes, total RNA from last larval instar silk glands was isolated using TRIzol reagent (Invitrogen, Carlsbad, CA), followed by isolation of mRNA using Dynabeads Oligo (dT)25 mRNA Purification Kit (Thermo Fisher Scientific, Waltham, USA), and cDNA was prepared using the NEXTflex Rapid RNA-Seq Kit (Bioo Scientific, Austin, USA). Additionally, to annotate the genome, RNA from heads, thoraces, and gonads of three male and female imagoes was extracted with TRI-Reagent (Sigma-Aldrich) according to the provided protocol. Biological replicas were pooled prior to isolation, resulting in three tissue-specific samples per each sex.

**Genome assembly and annotation.** To assemble the genome of *Y. cagnagella*, Oxford Nanopore reads were sequenced on the Nanopore PromethION platform by Novogene (HK) Co, Ltd. (Hong Kong, China). In addition, an Illumina library with 700 bp insert size was prepared and sequenced by the Genomics Core Facility of the European Molecular Biology Laboratory (Heidelberg, Germany) on the Illumina HiSeq 2500 with 250 bp paired-end reads. The raw reads were deposited in NCBI under SRA accession numbers SRR15714088 and SRR15714089.

First, adaptor sequences and low quality bases were filtered out of the Illumina data using Trimmomatic (version 0.36;[64]) with the following parameters: "ILLUMINACLIP: /PATH/TruSeq3-PE-2.fa:2:30:10:1:true SLIDINGWINDOW:4:20 MINLEN:100" and the quality of reads was inspected with FastQC (version 0.11.5;[65]). Genome size and heterozygosity was estimated from the filtered data using GenomeScope (version 1.0;[35]). K-mers of length 31 were counted by jellyfish (version 2.3.0;[66]).

Nanopore reads shorter than 500 bp and with a quality score lower than 7 were removed from the dataset with NanoFilt (version 2.7.1;[67]) and the reads were visualized using NanoPlot (version 1.33.1;[59]). Next, the FM-index Long Read Corrector (FMLRC version 1.0.0;[41]) was used with default settings to correct the long reads using the filtered Illumina sequences. As recommended, ropebwt2[68] and fmlrc-convert were used to construct the multi-string BWT data structure required by the FMLRC pipeline. The preprocessed long reads were then assembled with Flye (version 2.8;[69]) with settings adjusted to corrected input, *Y. cagnagella* genome size and three polishing iterations ("--nano-corr --genome-size 750 m --iterations 3").

To eliminate the haplotypic duplications from the primary assembly, purge_dups pipeline (version 1.0.1;[70]) was applied, followed by polishing using POLCA (MaSuRCA version 3.4.2;[42]). Quality assessment of the draft genome was performed using QUAST (version 4.6.3[71]), BUSCO tool suite (version 5.2.2, the dataset for the Insecta lineage;[72]) and the final genome was checked for contamination with Kraken 2 (version 1.0;[40]).

Repeat composition and average GC content were analyzed with RepeatModeler (version 1.0 (ref.[39], 2008-2015) and RepeatMasker (version 4.0[39]) software packages. To achieve more accurate masking, major satellites (Tab S3) were identified with TAREAN (version 0.3.8-451;[73]) from Illumina data subsampled to 0.25× genome coverage. Custom repeat library built from the genome sequence with RepeatModeler with added satellite dimers was used in

RepeatMasker pipeline to survey the landscape of repetitive elements and generate masked version of the *Y. cagnagella* assembly.

For genome annotation, all RNA-seq data (SRA accession numbers SRX17830525-SRX17830530) were concatenated into a single dataset, including the silk gland RNA-seq (see below). The quality of the sequencing was verified using FastQC (version 0.11.5;[65]). Resulting 2.87 Gb of sequence data were aligned to the masked genome assembly using STAR (version 2.7.7a;[74]). The genome index was generated with the following parameter scaled down to the size of *Y. cagnagella* genome: "--genomeSAindexNbases 13". Genes were predicted with BRAKER (version 2.1.5;[75]) and annotated using BLASTp with NCBI RefSeq invertebrate protein database[76], all implemented in the GenSAS platform (version 6.0;[77]). Gene set completeness was assessed by BUSCO (version 5.2.2;[72]) with the insecta_odb10 dataset and descriptive summaries were computed using gff3_stats.py script from the GenomeGC[78] suite.

Finally, the quality of the *Y. cagnagella* assembly was compared with summary statistics of other lepidopteran assemblies based on Oxford Nanopore data, namely the arched hooktip *Drepana arcuata*[79], the cocoa moth *Ephestia elutella*[80], the bagworm *Eumeta variegata*[2], the tobacco hornworm *Manduca sexta*[81], the meadow brown *Maniola jurtina*[82], the Mediterranean corn borer *Sesamia nonagrioides*[83] and the beet armyworm *Spodoptera exigua*[84], as well as with a reference genome of *Bombyx mori*[85] and genome of the close relative *P. xylostella*[86] (Table 1).

**Silk glands transcriptome preparation and analysis.** RNA isolation, synthesis of cDNA libraries and RNA sequencing were done as previously described[7] (see the section "Transcription specificity of silk candidate proteins and similarity between Y. cagnagella and Y. 200 evonymella candidate silk genes"). The cDNA library was sequenced on Illumina platform 2×150 bp (paired-end reads) with MiSeq. Adaptor sequences removal and trimming was performed using BBDUK (BBtools suite) with the following settings: ordered = t; ktrim = r; k = 23; mink = 11; hdist = 1; qtrim = rl; trimq = 20; min.length = 35; tpe; tbo. A further rRNA contamination step was conducted using BBDUK with the associated ribokmers.fa file[87], to eliminate rRNA contamination from the mRNA enrichment step of library preparation. Cleaned reads were assembled into a transcriptome using the multi k-mer rnaSPAdes assembler (version 3.13.1;[88]). K-mer sizes of 25, 35, 45, 55, 65 and 75 were chosen for *de novo* assembly to increase the likelihood of maximum transcript recovery[89]. Salmon (version 1.0.0;[90]) was used to quantify the abundance of transcripts. Transcripts with a TPM value of <1 were deemed as artefactual and thus removed from the final assembly. BUSCO (version 5.2.2;[72,91]) was used to assess the completeness of the assembly, using the Insecta odb10 dataset (https://busco.ezlab.org/). The raw reads used to generate the assembly are available via NCBI (SRA accession number: SRR15714087).

For sequence comparison, we used RNA sequencing data of *Y. evonymella* larvae from the BioProject PRJNA788289. The raw Illumina reads were processed and assembled following Yoshido et al.[92].

**Transcriptome annotation, ORF and signal peptide prediction.** The transcriptome was annotated using DIAMOND protein aligner (version 0.9.27.128;;[93]) with the latest NCBI Non-Redundant (nr) and Uniprot-Swissprot versions (accessed 26/11/19). The BlastX function was used with default settings and an e-value of $1.0 \times 10^{-5}$. Blast hits were then sorted based on bitscore, e-value, and percent identity. The best hit was retained for each transcript. ORFs were predicted using the Transdecoder. LongOrfs script from Transdecoder (version 5.5;[94]). ORFs were then searched for signal peptides using SignalP5.0b in eukaryote mode[95].

**Northern blotting, qPCR and sequence similarity to Y. evonymella.** Northern blotting was performed as previously described[16]. Tissues of *Y. cagnagella*: silk glands, larval carcasses with ablated silk glands and pupae, *Y. evonymella* silk glands and larval carcasses were homogenized and total RNA was isolated using RNA blue (Top-Bio, Vestec, Czech Republic). 5 µg of total RNA was separated on an agarose gel and blotted onto a nylon membrane. Probes were prepared by PCR, labeled with [α-$^{32}$P]dATP, and membranes were then hybridized and autoradiographed as described[16]. List of primers is shown in Table S1.

Four types of *Y. cagnagella* tissues (silk glands, gut, fat body and integument) in three biological replicates were analyzed by qPCR. Primers (Table S2) were designed with the software Lasergene PrimerSelect (DNASTAR, Madison, USA) to achieve the optimal function. The PCR reaction volume was 20 µl, and it contained 0.1 µg of diluted cDNA, 250 nM primers, and 4 µl of mixture HOT FIREPol EvaGreen qPCR Mix Plus (Solis BioDyne, Tartu, Estonia). After the initial denaturation/Pol activation step (95 °C for 15 min), 45 cycles (95 °C for 15 s; annealing temperature adjusted to the primer pair for 30 s; 72 °C for 20 s) were carried out using the instrument Rotor-Gene Q MDx 2plex HRM (Qiagen, Hilden, Germany). Each sample was analyzed in three technical replicates. The data were processed by the $2^{\Delta Ct}$ calculation using *elongation factor 1-α* as normalizer. The statistical significance was determined by Kruskal-Wallis rank sum test and Wilcoxon rank sum test.

For calculation of genetic differences between *Y. cagnagella* and *Y. evonymella* we used cDNAs encoding eight proteins detected in silk. Alignment and calculations of genetic differences (p-distance) were performed using MEGA-X software[96]. Only coding regions were used for the analysis without gaps and stop codons.

**Protein digestion, nLC-MS 2 analysis, and proteomic data search**. Protein samples were prepared for mass spectrometry as previously described[7]. Silk cocoon samples (5 mg) were dissolved in 8 M urea, trypsinized and acidified with trifluoroacetic acid (to 1% final concentration). The peptides were desalted and analyzed using nanoscale liquid chromatography coupled to tandem mass spectrometry (nLC-MS/MS). Peptides were analyzed and quantified using MaxQuant algorithms (version 1.5.3.8)[97]. The false discovery rate (FDR) was set at 1% for both proteins and peptides. The Andromeda search engine integrated into MaxQuant[98] was used to identify peptides by searching MS/MS spectra against a database derived from the transcriptome described above.

**Reporting summary**. Further information on research design is available in the Nature Portfolio Reporting Summary linked to this article.

## Data availability

The experimental data supporting the results of this study are available in this article or in the supplementary materials. The raw data have been deposited in NCBI under the bioproject accession numbers PRJNA760528 and PRJNA788289. The transcriptome and soft-masked genome assembly were deposited in the Dryad repository (https://doi.org/10.5061/dryad.4j0zpc8d3). For genome annotation, all RNA-seq data (SRA accession numbers SRX17830525-SRX17830530) were concatenated. List of silk gene candidates their GenBank accession codes are shown in (Table 2). The data underlying the graphs in Fig. 4 are available as Supplementary Data (Table S6). Uncropped and unedited blot/gel images are included as Supplementary Figure S12.

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

## Acknowledgements

This research was supported by European Community's Program Interreg Bayern Tschechische Republik Ziel ETZ 2021–2022 no. 331. The *Y. cagnagella* genome sequencing was funded by the grant 20-20650Y of the Czech Science Foundation awarded to P.N. This publication is also supported by the project "BIOCEV – Biotechnology and Biomedicine Centre of the Academy of Sciences and Charles University" (CZ.1.05/1.1.00/02.0109), from the European Regional Development Fund. We also acknowledge the core facility Laboratory of Electron Microscopy, Biology Centre CAS supported by the MEYS CR (LM2018129 Czech-BioImaging) and ERDF (No. CZ.02.1.01/0.0/0.0/16_013/0001775). M.H. was supported by IMG institutional support (RVO–68378050). We thank to Mrs. Jitka Pflegerová for her help with sample preparation.

## Author contributions

A.V., P.N., and Mi.Z. conceptualized the work, developed the methodology, and designed experiments. A.V., P.K., J.W. performed the analysis of *Y. cagnagella* genome, P.R. and I.P. collected insect material, P.D. and Mi.Z. analyzed transcriptome, H.S. performed histochemistry and electron microscopy. B.K. and L.R. performed transcriptional analysis, M.H. constructed cDNA libraries, M.S. and Ma.Z. performent phylogenetic analysis, P.N. and Mi.Z. wrote the manuscript with input from all authors.

## Competing interests

The authors declare no competing interests.
