## [Peer Review File · Communications Biology]

Reviewers' comments:

Reviewer #1 (Remarks to the Author):

This study has performed a multiple omics approach to thoroughly investigate “*Yponomeuta cagnagella*”, curated genes related to silk proteins, and opened up the molecular biology in Lepidoptera. The expansion of molecular information is essential in the field of ecology, and this paper will make a great contribution. In particular, the discussion on the identification of silk-related genes based on their expression specificity using Northern blotting and qPCR is highly appreciated. On the other hand, although the key point to make this paper more than just a “genome paper” is the discussion on evolution, the discussion was not sufficient.

Major comments:

- Introduction, the end of introduction

“In the present study, we show a detailed analysis of silk genes and their products in a closely related spindle ermine moth, *Yponomeuta cagnagella*. To identify and characterize all genes encoding major silk components, we sequenced and assembled the genome of *Y. cagnagella* and combined the results with transcriptomic and proteomic analyses.”

The scientific background of the focus on *Yponomeuta cagnagella* is unclear; it is not clear why new information on *Y. cagnagella* is needed when the main fibroin of *Yponomeuta* has already been reported (Yonemura and Sehnal, 2006). For the purpose of discussing evolution, I could not find a rational story to focus only on *Y. cagnagella*, even though *Yponomeuta* genus has more than 100 members.

- Abstract

“This study fills an important gap in our growing understanding of the structure, evolution, and function of silk genes and provides genomic resources for future studies of the chemical ecology of *Yponomeuta* species.”

- Discussion, the final paragraph

“This study fills an important gap in our growing understanding of the structure, evolution and function of silk genes and lays the foundation for future detailed comparative studies.”

While the authors claim to have discussed evolution, the analysis was not sufficient. There should be more to discuss than just sequence and gene expression similarities between closely related species. For example, there is now a large amount of sequence data in Lepidoptera, as the author lists in Table 1. I would like the author to do some more analysis on the relationship between the protein sequence I read in this article and other silk-related proteins in Lepidoptera.

Minor comments:

- Please number the lines.

- Introduction, the end of the first paragraph

“However, the increasing application of genomic technologies to characterize the nucleotide and protein sequences that form silk is revealing striking variability in silk properties across arthropod taxa.”

This sentence should cite the appropriate literatures.

Darwin's bark spider shares a spidroin repertoire with *Caerostris extrusa* but achieves extraordinary silk toughness through gene expression.

Kono N, Ohtoshi R, Malay AD, Mori M, Masunaga H, Yoshida Y, Nakamura H, Numata K, Arakawa K.

Open Biol. 2021 Dec;11(12):210242. doi: 10.1098/rsob.210242. Epub 2021 Dec 22.

PMID: 34932907

Multicomponent nature underlies the extraordinary mechanical properties of spider dragline silk.

Kono N, Nakamura H, Mori M, Yoshida Y, Ohtoshi R, Malay AD, Pedrazzoli Moran DA, Tomita M, Numata K, Arakawa K.

Proc Natl Acad Sci U S A. 2021 Aug 3;118(31):e2107065118. doi: 10.1073/pnas.2107065118.

PMID: 34312234

Annotated Draft Genomes of Two Caddisfly Species *Plectrocnemia conspersa* CURTIS and *Hydropsyche tenuis* NAVAS (Insecta: Trichoptera).
Heckenhauer J, Frandsen PB, Gupta DK, Paule J, Prost S, Schell T, Schneider JV, Stewart RJ, Pauls SU.
Genome Biol Evol. 2019 Dec 1;11(12):3445-3451. doi: 10.1093/gbe/evz264.
PMID: 31774498

Orb-weaving spider *Araneus ventricosus* genome elucidates the spidroin gene catalogue.
Kono N, Nakamura H, Ohtoshi R, Moran DAP, Shinohara A, Yoshida Y, Fujiwara M, Mori M, Tomita M, Arakawa K.
Sci Rep. 2019 Jun 10;9(1):8380. doi: 10.1038/s41598-019-44775-2.
PMID: 31182776

The bagworm genome reveals a unique fibroin gene that provides high tensile strength.
Kono N, Nakamura H, Ohtoshi R, Tomita M, Numata K, Arakawa K.
Commun Biol. 2019 Apr 29;2:148. doi: 10.1038/s42003-019-0412-8. eCollection 2019.
PMID: 31044173

The genome of an underwater architect, the caddisfly *Stenopsyche tienmushanensis* Hwang (Insecta: Trichoptera).
Luo S, Tang M, Frandsen PB, Stewart RJ, Zhou X.
Gigascience. 2018 Dec 1;7(12):giy143. doi: 10.1093/gigascience/giy143.
PMID: 30476205

The *Nephila clavipes* genome highlights the diversity of spider silk genes and their complex expression.
Babb PL, Lahens NF, Correa-Garhwal SM, Nicholson DN, Kim EJ, Hogenesch JB, Kuntner M, Higgins L, Hayashi CY, Agnarsson I, Voight BF.
Nat Genet. 2017 Jun;49(6):895-903. doi: 10.1038/ng.3852. Epub 2017 May 1.
PMID: 28459453

- Introduction, the second paragraph
"the fibroins, and the more soluble sheath proteins"
I recommend the author revises this sentence like this. "fibroins, and more soluble sheath proteins"

- Introduction, the second paragraph
"The coating proteins show a great heterogeneity between species, both in the number of proteins they contain and in the sequences of the individual components."
This sentence should cite the appropriate literatures.

- Materials and Methods, 2.2, the end of the paragraph
"using a Jeol JSM-7401F scanning electron microscope (Joel, Akishima, Japan)."
Not "Joel", "Jeol".

- Results, 3.1
The Fig S1 and Sig S2 numbers may be wrong, so check them carefully again.

- Results, 3.4, the final paragraph
"Finally, we estimated the evolutionary divergence and similarity between DNA and amino acid sequences by comparing the homologs of eight proteins and their cDNAs detected in the silks of *Y. cagnagella* and *Y. evonymella*. A summary can be found in Table S4 and shows 96-99% identity between the sequences of the two species. This supports the idea that individual silk components can be easily identified between the two species from the same genus based on similarities."
We need outgroups as a control.

- Results, 3.5, the third & fourth paragraphs

"The central part of Fib-H molecule consists of an imperfect repetitive sequence encoding 27-31 aa long repeat motifs."

It is easier to understand if authors represent it in a figure.

- Results, 3.5, the third & fourth paragraphs

"In addition, sericin 1 contains a sequence encoding a characteristic CVCY motif located 17 amino acid residues away from the C-terminus. A similar motif is also found in the *B. mori* and *G. mellonella* sericin 1 homologs."

It is unclear how it is distinctive, so represent it with a figure.

- Discussion, the fifth paragraph

"The fibroin of *Y. cagnagella* contains SSAAA sequence motifs suggesting its structural relationship to the fibroins of the X-ray class III and to the fibers of *A. yamamai* or *G. mellonella* 69,70."

It would be good to have a figure that makes it easier to understand the motifs mentioned in the fibroin, sericin, and fibrohexamerin sequences; the location of SSAAA is not known, so it is difficult to know if it is related to *A. yamamai* or *G. mellonella*.

- Discussion, the fifth paragraph

"Interestingly, we did not find any homolog of such Fhx-like gene in *Y. cagnagella*."

Does this mean that there are no hexamers as seen in *B. mori* silk?

- Figure S2

Italicize the species name.

Reviewer #2 (Remarks to the Author):

1) Authors did not conduct ab initio gene prediction. To characterize this species, authors must perform gene prediction and compare its gene repertoire with other species.

2) In L335, "26,054 open reading frames (ORFs) were found in the assembly" is not enough. How many transcripts (not ORF) were identified in the assembly? I understand the purpose of this research (and authors did not need to care about non-coding transcripts), but it is important to state general information of the transcriptome.

3) In L334-335, can the one hundred forty six missing BUSCOs be found in the genome assembly? If so, authors must note that for readers to understand the reason for "missing BUSCOs" is merely not being expressed in SG.

4) The corresponding relations between MS and transcriptome analysis must be stated to guarantee the quality of MS and RNA-seq. Are one hundred twenty proteins detected by MS surely found in the transcriptome assembly?

5) I am afraid that illumina short read data can not correctly retrieve the whole sequence of highly repetitive sequence such as Fib-H gene. To correctly predict Fib-H gene sequence, authors must perform Iso-seq using PacBio or Nanopore (Kono et al., PNAS. 2021).

Reviewer #3 (Remarks to the Author):

The manuscript entitled "Genome sequence and silkomics of the spindle ermine moth, *Yponomeuta cagnagella*, representing the early-diverging lineage of the ditrysian Lepidoptera" provides draft genome and silk transcriptome information of the spindle ermine moth, *Y. cagnagella*. Volenikova et al. obtained high quality (long and accurate) draft genome by hybrid assembly using Oxford Nanopore and Illumina reads. The authors also obtained transcriptome assembly by sequencing larval silk grand cDNA. These genomic and transcriptomic data were properly deposited to public database. The authors specially focused on silk and the secretory organ, silk glands. They collected complete gene sequences of *Y. cagnagella* silk components from the transcriptome assembly and

verified the tissue specific expression using Northern blotting and qPCR analysis. They also characterized the morphology of *Y. cagnagella* silk glands and cocoon by light and electron microscopy, respectively. The results of this study are clear and informative for especially silk science and the manuscript is overall well written. However, it is difficult to understand the novelty and importance of this study even though the reviewer is standard insect scientist. Therefore, I recommend that the authors resubmit the manuscript to more suitable journal.

Major comments:

- Although the authors succeeded to obtain long and accurate *Y. cagnagella* genome assembly using Oxford Nanopore and Illumina reads, the methods and the assembly metrics are not special. The assembly information is new and very informative but novel and important results are not provided in this study.
- To provide more accurate transcript sequences of *Y. cagnagella* silk components, the authors should obtain information of the transcriptional start sites and splicing variants.
- The reviewer expects that the authors collect more RNA-seq data from various tissues and generate gene models based on the reference genome obtained in this study.

Reviewer #4 (Remarks to the Author):

Volenikova and colleagues perform comprehensive analysis to characterize the “silkome” of the spindle ermine moth. This is a great (and much needed!) work. I was impressed by the number of analyses performed by the researchers and they certainly contributed new important information to better understand silk gene/development evolution in Lepidoptera. My primary concerns have to do with the de novo assembly. These comments are organized below by line number. If the authors clarify a few of these issues, I would be more than happy to see this paper published.

44: A couple of other studies on caddisfly silk: Luo et al. 2018 (<https://doi.org/10.1093/gigascience/giy143>), Frandsen et al. 2019 (<https://doi.org/10.1098/rstb.2019.0206>).

73: what does colonize extensively mean?

147: Why multiple individuals? With any sort of heterozygosity among the individuals, this would negatively affect the assembly.

171: I would also suggest running BUSCO with the Endopterygota lineage dataset as you will be able to search for more lineage-specific genes.

208: Did you perform annotation only on the transcriptome? Or also on the genome? Might consider adding “Transcriptome” to the 2.7 heading.

274: 10.5% is still more duplicated genes than I would expect. I suspect this might have something to do with using four individuals for the nanopore sequencing and an additional individual for Illumina sequencing. How did you choose your coverage cutoffs for purge_dups? Did you look at the kmer histogram to set them manually? Often the defaults are not good enough. I'd also strongly recommend doing GenomeScope profiling to both visualize the kmer distribution and have another method for estimating genome size.

288: How did you estimate heterozygosity? If you used five different individuals in the sequencing, how are you sure that this represents true heterozygosity? Was it just estimated from the Illumina reads? I would suggest running genomescope on the Illumina reads to see if the value for genome-wide heterozygosity is similar to whatever method you've used here.

437: I wouldn't classify this genome as “comparable to current standards for non-model Lepidoptera species”. The current best standard for lepidopteran genome assemblies is coming from the Darwin Tree of Life Project. Nearly all of these genomes have contig N50s greater than 1

Mbp, with most greater than 10 Mbp (see Hotaling et al. 2021, <https://doi.org/10.1073/pnas.2109019118>, Fig. 5A). The authors should acknowledge the current state-of-the-art in genome assemblies that are currently being generated. This isn't to say that their conclusions on silk genes, etc. are not well-founded (on the contrary, I think the genome assembly that they produced here is suitable for their conclusions!), but it would be appropriate to acknowledge that much more contiguous genomes are being produced across non-model organisms in Lepidoptera.

The answers to reviewer's comments

Reviewer #1:

This study has performed a multiple omics approach to thoroughly investigate "*Yponomeuta cagnagella*", curated genes related to silk proteins, and opened up the molecular biology in Lepidoptera. The expansion of molecular information is essential in the field of ecology, and this paper will make a great contribution. **In particular, the discussion on the identification of silk-related genes based on their expression specificity using Northern blotting and qPCR is highly appreciated. On the other hand, although the key point to make this paper more than just a "genome paper" is the discussion on evolution, the discussion was not sufficient.**

Major comments:

(1) - Introduction, the end of introduction

"In the present study, we show a detailed analysis of silk genes and their products in a closely related spindle ermine moth, *Yponomeuta cagnagella*. To identify and characterize all genes encoding major silk components, we sequenced and assembled the genome of *Y. cagnagella* and combined the results with transcriptomic and proteomic analyses."

The scientific background of the focus on *Yponomeuta cagnagella* is unclear; it is not clear why new information on *Y. cagnagella* is needed when the main fibroin of *Yponomeuta* has already been reported (Yonemura and Sehnal, 2006). For the purpose of discussing evolution, I could not find a rational story to focus only on *Y. cagnagella*, even though *Yponomeuta* genus has more than 100 members.

Action taken

*We have analyzed silks in both *Y. cagnagella* and *Y. evonymella* independently using transcriptomics and proteomics, and by using both related species, we can conveniently add genes to the *Y. cagnagella* data that we could not directly detect for some reason but found homologs in *Y. evonymella*. In this study, we found three additional silk genes based on homology with *Y. evonymella*.*

(2) Abstract

"This study fills an important gap in our growing understanding of the structure, evolution, and function of silk genes and provides genomic resources for future studies of the chemical ecology of *Yponomeuta* species."

- Discussion, the final paragraph

"This study fills an important gap in our growing understanding of the structure, evolution and function of silk genes and lays the foundation for future detailed comparative studies."

While the authors claim to have discussed evolution, the analysis was not sufficient. There should be more to discuss than just sequence and gene expression similarities between closely related species. For example, there is now a large amount of sequence data in Lepidoptera, as the author lists in Table 1. I would like the author to do some more analysis on the relationship between the protein sequence I read in this article and other silk-related proteins in Lepidoptera.

Action taken

We have rewritten the discussion section and added the information on evolution of silk proteins as recommended.

(3) Minor comments:

- **Please number the lines.**

Action taken

The lines were numbered as suggested.

(4) - Introduction, the end of the first paragraph

“However, the increasing application of genomic technologies to characterize the nucleotide and protein sequences that form silk is revealing striking variability in silk properties across arthropod taxa.”

This sentence should cite the appropriate literatures.

Action taken

We have modified the introductory paragraph of this article to clarify that the silk arthropods are a product of convergent evolution and evolved independently in different groups. We have therefore added citations that clarify this issue and focus on the silks of Lepidoptera and Trichoptera. The silks of Lepidoptera and Trichoptera, which form the superorder Amphiesmenoptera, have a common origin.

(5) - Introduction, the second paragraph

“the fibroins, and the more soluble sheath proteins”

I recommend the author revises this sentence like this. “fibroins, and more soluble sheath proteins”

Action taken

We change the text (lines 58 and 59) to read: “insoluble fibroin proteins, which form two core filaments and sericin proteins soluble in hot water which form a hydrophilic coating and seal the filaments into a fiber”.

(6) - Introduction, the second paragraph

“The coating proteins show a great heterogeneity between species, both in the number of proteins they contain and in the sequences of the individual components.”

This sentence should cite the appropriate literatures.

Action taken

We added the citation as suggested (line 64).

(7) - Materials and Methods, 2.2, the end of the paragraph

“using a Jeol JSM-7401F scanning electron microscope (Joel, Akishima, Japan).”

Not “Joel”, “Jeol”.

Action taken

The error has been corrected in the main text.

(8) - Results, 3.1

The Fig S1 and Sig S2 numbers may be wrong, so check them carefully again.

Action taken

The error has been corrected in the main text.

(9) - Results, 3.4, the final paragraph

“Finally, we estimated the evolutionary divergence and similarity between DNA and amino acid sequences by comparing the homologs of eight proteins and their cDNAs detected in the silks of *Y. cagnagella* and *Y. evonymella*. A summary can be found in Table S4 and shows 96-99% identity between the sequences of the two species. This supports the idea that individual silk components can be easily identified between the two species from the same genus based on similarities.”

We need outgroups as a control.

Action taken

We added T. bisselliella sequences as the outgroup as suggested (it is now Table S5).

(10) - Results, 3.5, the third & fourth paragraphs

“The central part of Fib-H molecule consists of an imperfect repetitive sequence encoding 27-31 aa long repeat motifs.”

It is easier to understand if authors represent it in a figure.

Action taken

We added Figure S4 showing the H-fib repeats.

(11) - Results, 3.5, the third & fourth paragraphs

“In addition, sericin 1 contains a sequence encoding a characteristic CVCY motif located 17 amino acid residues away from the C-terminus. A similar motif is also found in the *B. mori* and *G. mellonella* sericin 1 homologs.”

It is unclear how it is distinctive, so represent it with a figure.

Action taken

We added Figure S9 showing the homology at the C-termini of sericin-1 sequences in several species.

(12) - Discussion, the fifth paragraph

“The fibroin of *Y. cagnagella* contains SSAAA sequence motifs suggesting its structural relationship to the fibroins of the X-ray class III and to the fibers of *A. yamamai* or *G. mellonella* 69,70.”

It would be good to have a figure that makes it easier to understand the motifs mentioned in the fibroin, sericin, and fibrohexamerin sequences; the location of SSAAA is not known, so it is difficult to know if it is related to *A. yamamai* or *G. mellonella*.

Action taken

We have added Figure S9 showing homology at the C termini of sericin-1 sequences in different species.

(13) - Discussion, the fifth paragraph

“Interestingly, we did not find any homolog of such Fhx-like gene in *Y. cagnagella*.”

Does this mean that there are no hexamers as seen in *B. mori* silk?

Action taken

We have adjusted the text (line 511) explaining that there are two separate silk protein families – true Fhx and Fhx-like proteins. The function of Fhx-like proteins is not clear.

(14) - Figure S2

Italicize the species name.

Action taken

We have corrected the text as suggested.

Reviewer #2 (Remarks to the Author):

(15) Authors did not conduct ab initio gene prediction. To characterize this species, authors must perform gene prediction and compare its gene repertoire with other species.

Action taken

*We performed the gene prediction as recommended and added corresponding text to Materials and methods, Results and Discussion. Regarding comparison with other species, we added a column in Tab1 with predicted gene number in *Y. cagnagella* and other lepidopteran species.*

(16) In L335, "26,054 open reading frames (ORFs) were found in the assembly" is not enough. **How many transcripts (not ORF) were identified in the assembly? I understand the purpose of this research (and authors did not need to care about non-coding transcripts), but it is important to state general information of the transcriptome.**

Action taken

We have added the information as suggested (line 255): „Our assembly contained 31152 transcripts”.

(17) In L334-335, can the one hundred forty six missing BUSCOs be found in the genome assembly? If so, authors must note that for readers to understand the reason for "missing BUSCOs" is merely not being expressed in SG.

Action taken

There were 146 BUSCO orthologs missing from the transcriptome and only three from the genome (most of the 143 genes are probably not expressed in SG). We have added the note (line 354): “We concluded that the 146 missing BUSCOs are likely the result of high SG specialisation for silk production”.

(18) The corresponding relations between MS and transcriptome analysis must be stated to guarantee the quality of MS and RNA-seq. **Are one hundred twenty proteins detected by MS surely found in the transcriptome assembly?**

Action taken

*The transcriptome is used to create a protein database for the detection of tryptic peptides by MS/MS. We obtained almost the same number of silk proteins in the silk of *Y. evonymella*, which were analyzed independently, which confirms the good quality of the data.*

(19) **I am afraid that illumina short read data cannot correctly retrieve the whole sequence of highly repetitive sequence such as Fib-H gene.** To correctly predict Fib-H gene sequence, authors must perform Iso-seq using PacBio or Nanopore (Kono et al., PNAS. 2021).

Action taken

As mentioned in the Abstract (Line 30) and Material and Methods (Line 164) we used Oxford Nanopore reads.

Reviewer #3 (Remarks to the Author):

The manuscript entitled “Genome sequence and silkomics of the spindle ermine moth, *Yponomeuta cagnagella*, representing the early-diverging lineage of the ditrysian Lepidoptera” provides draft genome and silk transcriptome information of the spindle ermine moth, *Y. cagnagella*. Volenikova et al. obtained high quality (long and accurate) draft genome by hybrid assembly using Oxford Nanopore and Illumina reads. The authors also obtained transcriptome assembly by sequencing larval silk grand cDNA. These genomic and transcriptomic data were properly deposited to public database. The authors specially focused on silk and the secretory organ, silk glands. They collected complete gene sequences of *Y. cagnagella* silk components from the transcriptome assembly and verified the tissue specific expression using Northern blotting and qPCR analysis. They also characterized the morphology of *Y. cagnagella* silk grands and cocoon by light and electron microscopy, respectively. The results of this study are clear and informative for especially silk science and the manuscript is overall well written. However, it is difficult to understand the novelty and importance of this study even though the reviewer is standard insect scientist. Therefore, I recommend that the authors resubmit the manuscript to more suitable journal.

Major comments:

(20) Although the authors succeeded to obtain long and accurate *Y. cagnagella* genome assembly using Oxford Nanopore and Illumina reads, the methods and the assembly metrics are not special. **The assembly information is new and very informative but novel and important results are not provided in this study.**

Action taken

Due to frequent gene loss, gene duplication, adaptive changes, and high divergence, it is virtually impossible to find homologous silk genes (except fibroins) via BLAST. We need to establish a high quality framework of well-characterised silk genes from different lepidopteran species that will serve for further characterization and gene annotation. The new data added to the supplemental information illustrates this in detail.

(21) • **To provide more accurate transcript sequences of *Y. cagnagella* silk components, the authors should obtain information of the transcriptional start sites and splicing variants.**

Action taken

Major focuss of our study is the detection of all major silk components. The transcriptional start sites and splicing variants will require separate study.

(22) **The reviewer expects that the authors collect more RNA-seq data from various tissues and generate gene models based on the reference genome obtained in this study.**

Action taken

We have performed genome annotation using additional RNA-seq data and incorporated it to the main text.

Reviewer #4 (Remarks to the Author):

Volenikova and colleagues perform comprehensive analysis to characterize the “silcome” of the spindle ermine moth. This is a great (and much needed!) work. I was impressed by the number of analyses performed by the researchers and they certainly contributed new important information to better understand silk gene/development evolution in Lepidoptera. My primary concerns have to do with the de novo assembly. These comments are organized below by line number. If the authors clarify a few of these issues, I would be more than happy to see this paper published.

(23) **A couple of other studies on caddisfly silk:** Luo et al. 2018 (<https://doi.org/10.1093/gigascience/giy143>), Frandsen et al. 2019 (<https://doi.org/10.1098/rstb.2019.0206>).

Action taken

We have added both citations to the main text (lines 45-46).

(24): **what does colonize extensively mean?**

Action taken

We have adjusted the text (line 76) as follows: “colonizing herbs, shrubs and trees on a large scale”.

(25): **Why multiple individuals?** With any sort of heterozygosity among the individuals, this would negatively affect the assembly.

Action taken

*Multiple individuals were used in order to obtain enough genomic DNA for OxfordNanopore sequencing, as the yield from a single *Y. cagnagella* pupa was unfortunately not sufficient for library preparation. To decrease the negative effect of added heterozygosity, specimen from the same family were used for the isolation.*

(26): **I would also suggest running BUSCO with the Endopterygota lineage dataset** as you will be able to search for more lineage-specific genes.

Action taken

We ran the BUSCO with the Endopterygota dataset as recommended. The results are shown below, confirming the quality of the assembly:

C:95.4%[S:85.9%,D:9.5%],F:2.7%,M:1.9%,n:2124

(27): **Did you perform annotation only on the transcriptome? Or also on the genome? Might consider adding “Transcriptome” to the 2.7 heading.**

Action taken

We added “Transcriptome” to the 2.7 heading in the main text. Genome annotation was performed as well and corresponding text was added to Materials and Methods, Results and Discussion.

(28): 10.5% is still more duplicated genes than I would expect. I suspect this might have something to

do with using four individuals for the nanopore sequencing and an additional individual for Illumina sequencing. **How did you choose your coverage cutoffs for purge_dups? Did you look at the kmer histogram to set them manually?** Often the defaults are not good enough. **I'd also strongly recommend doing GenomeScope profiling to both visualize the kmer distribution and have another method for estimating genome size.**

Action taken

We agree that 10.5% duplicated BUSCO orthologs is more than expected for a purged assembly and with high probability it is a consequence of using multiple specimens for gDNA isolation. The ideal cutoffs for purge_dups were estimated by the pipeline and manually verified, based on the read depth histogram (see below, Fig R1). Multiple manual adjustments of the values were tested, however we have not managed to purge the duplications further without also losing single copy BUSCO orthologs. The overall degree of deduplication was verified by K-mer Analysis Toolkit (version 2.3.4) and is shown on Fig R2.

GenomeScope visualization of the k-mer spectrum was performed and is shown on Fig. S3. We thank the reviewer for pointing out that it is not properly covered in the main text, as we only included it in the Supplements. Corresponding parts of methods and results were added to sections 2.5, 3.1 and 4.

Fig R1. Read depth histogram of *Y. cagnagella* genome assembly with cutoff values used for the purge_dups pipeline indicated by vertical lines.

Fig R2. Comparison of k-mer multiplicity in Illumina reads and genome assembly before and after purging showed major reduction of multiplicates

(29): How did you estimate heterozygosity? If you used five different individuals in the sequencing, how are you sure that this represents true heterozygosity? Was it just estimated from the Illumina reads? I would suggest running genomescope on the Illumina reads to see if the value for genome-wide heterozygosity is similar to whatever method you've used here.

Action taken

We used GenomeScope profiling on the Illumina reads as suggested by the Reviewer (results shown in Fig S3).

(30) I wouldn't classify this genome as "comparable to current standards for non-model Lepidoptera species". The current best standard for lepidopteran genome assemblies is coming from the Darwin Tree of Life Project. Nearly all of these genomes have contig N50s greater than 1 Mbp, with most greater than 10 Mbp (see Hotaling et al. 2021, <https://doi.org/10.1073/pnas.2109019118>, Fig. 5A). The authors should acknowledge the current state-of-the-art in genome assemblies that are currently being generated. This isn't to say that their conclusions on silk genes, etc. are not well-founded (on the contrary, I think the genome assembly that they produced here is suitable for their conclusions!), but it would be appropriate to acknowledge that much more contiguous genomes are being produced across non-model organisms in Lepidoptera.

Action taken

We agree with the Reviewer and adjusted the main text accordingly (lines 474-477).

REVIEWERS' COMMENTS:

Reviewer #1 (Remarks to the Author):

Appropriate corrections were identified.
There are no problem to accept.

Reviewer #2 (Remarks to the Author):

The points I raised in the previous reveiw, have been addressed in good faith and I am satisfied with the authors' corrections.